# Reform influences location of death: Interrupted time-series analysis on older adults and persons with dementia

Janet L. MacNeil Vroomen[1]☯*, Camilla Kjellstadli[2]☯, Heather G. Allore[3,4], Jenny T. van der Steen[5,6], Bettina Husebo[2,7]

1 Department of Internal Medicine, Section of Geriatric Medicine, Amsterdam Public Health Research Institute, Amsterdam University Medical Center, University of Amsterdam, Amsterdam, North Holland, The Netherlands, 2 Department of Global Public Health and Primary Care, University of Bergen, Bergen, Norway, 3 Department of Internal Medicine, Section of Geriatrics, School of Medicine, Yale University, New Haven, Connecticut, The United States of America, 4 Department of Biostatistics, Yale School of Public Health, New Haven, Connecticut, The United States of America, 5 Department of Public Health and Primary Care, Leiden University Medical Center, Leiden, The Netherlands, 6 Department of Primary and Community Care, Radboud University Medical Center, Nijmegen, The Netherlands, 7 Municipality of Bergen, Bergen, Norway

☯ These authors contributed equally to this work.
* j.l.macneil-vroomen@amsterdammumc.nl

**Data Availability Statement:** All relevant data are within the manuscript and its Supporting Information files.

## Abstract

### Background

Norway instituted a Coordination Reform in 2012 aimed at maximizing time at home by providing in-home care through community services. Dying in a hospital can be highly stressful for patients and families. Persons with dementia are particularly vulnerable to negative outcomes in hospital. This study aims to describe changes in the proportion of older adults with and without dementia dying in nursing homes, home, hospital and other locations over an 11-year period covering the reform.

### Methods and findings

This is a repeated cross-sectional, population-level study using mortality data from the Norwegian Cause of Death Registry hosted by the Norwegian Institute of Public Health. Participants were Norwegian older adults 65 years or older with and without dementia who died from 2006 to 2017. The policy intervention was the 2012 Coordination Reform that increased care infrastructure into communities. The primary outcome was location of death listed as a nursing home, home, hospital or other location. The trend in the proportion of location of death, before and after the reform was estimated using an interrupted time-series analysis. All analyses were adjusted for sex and seasonality. Of the 417,862 older adult decedents, 61,940 (14.8%) had dementia identified on their death certificate. Nursing home deaths increased over time while hospital deaths decreased for the total population (adjusted Relative Risk Ratio (aRRR) 0.87, 95% CI 0.82–0.92) and persons with dementia (aRRR: 0.93, 95%CI 0.91–0.96) after reform implementation.

**Funding:** JMV received a ZonMw career award, Veni grant # 91619060. https://www.zonmw.nl/nl/onderzoek-resultaten/fundamenteel-onderzoek/programmas/programma-detail/veni/t/commissie-13/. The funders had no role in study design, data collection and analysis, decision to publish, or preparation of the manuscript.

**Competing interests:** NO authors have competing interests.

## Conclusion

This study provides evidence that the 2012 Coordination Reform was associated with decreased older adults dying in hospital and increased nursing home death; however, the number of people dying at home did not change.

## Introduction

After the Coordination Reform was introduced in a white paper in 2009, the Norwegian government implemented it in 2012 as a response to increasing costs, to ensure sustainability of the health care system [1]. Through administrative, structural and economic approaches, tasks and responsibility were transferred from secondary to primary care with a goal to decrease hospitalizations and ensure that services were provided at the lowest efficient care level, closer to the patient's home [1–3]. Municipalities became responsible for caring for patients discharged from hospital after a shorter length of stay. After the reform, length of hospitalizations decreased for older adults, re-admission rates increased, and discharge rates to short-term nursing homes (NHs) from hospitals increased [3, 4]. The reform included opening of municipal emergency bed units, which led to reduced hospitalizations for some conditions [3, 5]. A study from one Norwegian nursing home found a 15% increase in mortality for older adults discharged from hospital to the nursing home post reform [3]. It is unknown if these results can be found at the national level.

Norway has universal healthcare. Municipalities are responsible for primary care, offering home nursing services, short-term and long-term NH care, rehabilitation, and ensure access to a general practitioner and out-of-hours services. The government is responsible for secondary care including hospitals. Short-term NH beds have increased after the reform, at the expense of long-term NH beds [6, 7]. More emphasis was added to providing care to persons at home to allow them to stay longer at home [8]. It is unknown whether place of death was affected.

It is also unclear how people with dementia were affected, arguably the most vulnerable and costliest patient population. Dementia is a chronic debilitating condition with cognitive, behavior and functional decline, and a life span varying from 3–12 years from diagnosis [9, 10]. In Western countries most persons with dementia die in NHs [11] and in Norway, this figure is particularly high (93%) [12, 13]. The rationale for institutionalization may differ for persons with dementia compared to persons without dementia because informal caregiver stress in addition to patient characteristics are predictors of institutionalization [14].

It is also unknown if the Coordination Reform is associated with more persons with dementia dying in hospital. Persons with dementia admitted to hospital are at risk for functional decline, lack of pain control, increased morbidity, increased mortality and a decreased quality of life [15–17]. Furthermore, numerous nursing and medical procedures may be unnecessarily continued or started in the last hours of a patient with dementia's life [18] and persons with dementia are at high risk for delirium [19].

There is limited health policy research to determine whether the reform has been effective. The aim of this study is to assess the impact of the 2012 Coordination Reform on location of death for the total population older than 65 years with and without dementia, based on population-level data. We hypothesize the reform would be associated with a gradual increase in the proportion of people dying in NHs and at home and a decrease in the proportion dying in hospital for persons without dementia. For persons with dementia, we do not expect a difference in the proportions that die in a NH setting because of caregiver burden and as this reform

involves not only building infrastructure but an awareness and readiness for change, more time may be required to see a difference in location of death for this population.

## Methods

### Study design

We performed an interrupted time series analysis (ITS) based on guidelines by Bernal et al. [20] to determine whether the introduction of the Coordination Reform of January 1, 2012 was associated with changes in location of death for the total Norwegian population ≥65 years and for persons with and without dementia. In an ITS study, a time series of the outcome of interest is used to establish an underlying trend, which is 'interrupted' by an intervention at a known point in time [20]. The pre-intervention underlying trend is compared against the post-intervention period to identify whether the intervention is associated with changes in the outcome [20]. Interrupted time series is increasingly used to evaluate policy in public health [20].

We used repeated cross-sectional, open-access national-level aggregated data on location of death from the Norwegian Cause of Death Registry (NCoDR), providing 100% coverage of the Norwegian population. We used quarterly data spanning the period January 1, 2006 through December 31, 2017. This study was developed using the STROBE and RECORD statement guidelines (S1 Table) [21]. As the data was public and anonymized, ethics approval was not required.

### Participants

Individuals included were 65 years and older at the time of death. Decedents´ location of death was recorded as at home, in a hospital, a NH, or other setting (abroad, under transportation to hospital, other specified). Statistics Norway provided total number of older adults that died per quarter and per location of death. Persons with dementia were identified based on International Classification of Diseases, Tenth Revision (codes for dementia: F00.0, F00.1, F00.2, F00.9, F01.0, F01.1, F01.2, F01.3, F01.8, F01.9, F02.0, F02.1, F02.2, F02.3, F02.4, F02.8, F03, G30.0, G30.1, G30.8, G30.9) if dementia diagnosis was included as one of the diagnoses anywhere on the death certificate.

### Variables

**Primary outcome.** Place of death as recorded on the death certificate were categorized into home, NH, hospital and other (specified).

**Explanatory variables.** A time variable (in cumulative quarters) and policy dummy variables indicating the pre-intervention period (coded 0) or the post-intervention period (coded 1) were created. Calendar quarters were included as a categorical variable in the model to account for seasonality [20]. Sex was included as a covariate.

### Statistical analysis

Study population characteristics and the distribution of place of death were described using unadjusted proportions. Summaries and bivariate comparisons between the outcomes and potential time-varying confounders, and basic before-and-after comparisons were performed [20].

Three weighted multinomial logistic regressions were performed for the total population, persons with dementia and without to calculate adjusted relative risk ratios (aRRR) and year-specific mean predicted probabilities of location of death. When calculating predicted

probabilities, all other variables were held at their means. Death in nursing home was the reference group. The regression analyses were weighted to adjust for population growth over the study period. Models included time in cumulative quarters since the start of the study, a reform variable, and an interaction term between the reform and cumulative quarters variable. The cumulative quarters variable can be interpreted as the quarterly aRRR of dying in a particular location pre-reform. The reform variable can be interpreted as the immediate (step) change following the implementation of the reform. The interaction between the cumulative quarters and the reform variables can be interpreted as the quarterly change in relative risk of dying at a particular location since the introduction of the reform (slope change). Lag variables were not created because the policy was enacted on January 1, 2012 after 3 years of notice. There were economic sanctions for municipalities who were unprepared before January 1, 2012 [1, 4]. Calendar quarters were included as a categorical variable in the model to account for seasonality [20]. Stata version 16 was used for all analyses.

## Results

Table 1 shows unadjusted yearly locations of death proportions (2006–2017) per study population.

### Multinomial logistic regressions

**Total population ≥65 years.** Before the introduction of the 2012 reform, the proportions of people dying at home (Cumulative quarters, aRRR 0.97, 95% CI 0.96–0.98), hospital (aRRR 0.95, 95% CI 0.95–0.96) and elsewhere (aRRR 0.94, 95% CI 0.93–0.95) were significantly decreasing compared to NHs (Fig 1, S2 Table). After the introduction of the reform, there was evidence of a step change in the proportion of home deaths (aRRR 1.19, 95%CI 1.10–1.29) while the proportion of hospital deaths (aRRR 0.87 95%CI 0.82–0.92) and dying elsewhere (aRRR 0.71, 95%CI 0.62–0.82) decreased compared to NH deaths. This was followed by a small but significant deceleration (e.g. reduced slope) in home deaths (aRRR 0.98, 95%CI

**Table 1. Location of death (2006–2017) for all Norwegian adults over 65 years by dementia status (%).**

| Years | N | Total population Nursing Home (%) | Home (%) | Hospital (%) | Elsewhere (%) | N | Dementia Nursing Home (%) | Home (%) | Hospital (%) | Elsewhere (%) | N | No Dementia Nursing Home (%) | Home (%) | Hospital (%) | Elsewhere (%) |
|---|---|---|---|---|---|---|---|---|---|---|---|---|---|---|---|
| 2006 | 34,631 | 46.5 | 12.6 | 37.2 | 3.7 | 4,205 | 84.9 | 4.4 | 9.0 | 1.6 | 30,426 | 41.2 | 13.7 | 41.1 | 4.0 |
| 2007 | 35,478 | 46.8 | 12.8 | 36.4 | 4.0 | 4,434 | 85.3 | 5.1 | 8.7 | 1.0 | 31,044 | 41.3 | 13.9 | 40.4 | 4.4 |
| 2008 | 34,898 | 48.3 | 12.3 | 36.2 | 3.3 | 4,396 | 84.6 | 4.7 | 9.4 | 1.2 | 30,502 | 43.1 | 13.4 | 40 | 3.6 |
| 2009 | 34,445 | 49.5 | 12.1 | 35.1 | 3.3 | 4,502 | 86.7 | 4.4 | 7.5 | 1.5 | 29,943 | 43.9 | 13.3 | 39.3 | 3.5 |
| 2010 | 34,628 | 50.4 | 12.7 | 33.9 | 3.0 | 4,862 | 86.8 | 4.5 | 7.9 | 0.7 | 29,766 | 44.5 | 14.1 | 38.1 | 3.4 |
| 2011 | 34,587 | 51.7 | 12.1 | 32.8 | 3.4 | 5,009 | 87.5 | 4.3 | 6.7 | 1.5 | 29,578 | 45.6 | 13.5 | 37.3 | 3.7 |
| 2012[a] | 35,457 | 53.5 | 12.6 | 31.1 | 2.8 | 5,307 | 86.9 | 5.1 | 6.9 | 1.1 | 30,150 | 47.6 | 14 | 35.3 | 3.1 |
| 2013[a] | 34,764 | 53.5 | 12.4 | 30.7 | 3.4 | 5,272 | 87.1 | 4.8 | 6.7 | 1.5 | 29,492 | 47.5 | 13.7 | 35 | 3.8 |
| 2014[a] | 34,297 | 53.9 | 12.2 | 30.5 | 3.3 | 5,569 | 86.6 | 5.4 | 6.8 | 1.2 | 28,728 | 47.6 | 13.5 | 35.2 | 3.7 |
| 2015[a] | 34,859 | 54.2 | 12.1 | 30.5 | 3.2 | 5,820 | 87.0 | 4.9 | 7.0 | 1.1 | 29,039 | 47.6 | 13.6 | 35.2 | 4 |
| 2016[a] | 34,724 | 56.6 | 11.1 | 28.9 | 3.3 | 6,141 | 89.1 | 2.8 | 6.9 | 1.3 | 28,583 | 49.7 | 12.9 | 33.7 | 4.4 |
| 2017[a] | 35,094 | 57.3 | 10.6 | 28.9 | 3.2 | 6,423 | 90.5 | 2.6 | 6.0 | 0.9 | 28,671 | 49.9 | 12.4 | 34 | 3.6 |
| Total | 417,862 | | | | | 61,940 | | | | | 355,922 | | | | |

[a] Indicates post Coordination Reform.

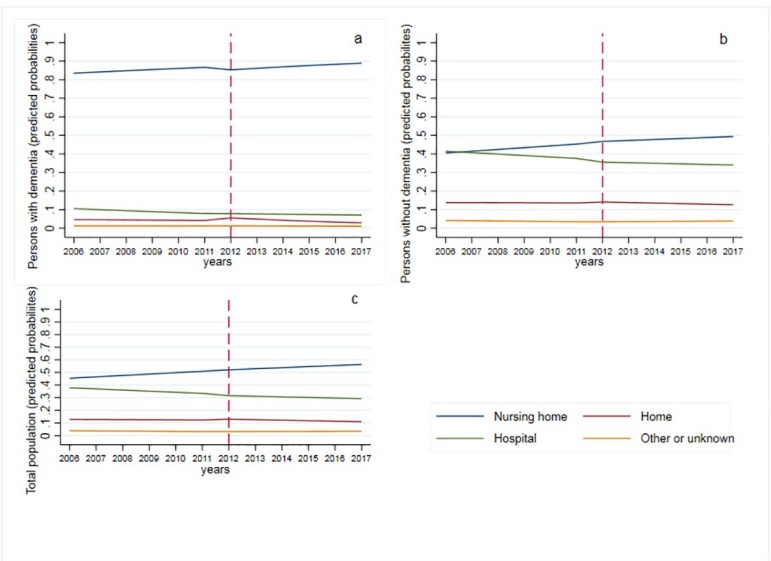

**Fig 1. Location of death for persons with dementia (panel a) without dementia (panel b) and the total population (panel c), from multinomial logistic regression (predicted probabilities (y-axis) plotted over time) weighted to adjust for population growth over the study period.** Note red dashed line is the implementation of the 2012 long term care reform.

0.96–0.98) and a similarly small but significant acceleration (e.g. increased slope) in hospital deaths (aRRR 1.02, 95%CI 1.01–1.02) and dying elsewhere (aRRR 1.06, 95%CI 1.05–1.08) in recent years compared to previous years. Males were more likely than females to die at home (aRRR 1.86, 95%CI 1.82–1.90), in hospital (aRRR 1.75, 95%CI 1.73–1.78) or in another location (aRRR 2.53, 95%CI 2.44–2.62) than in a NH.

**Persons without dementia.** Pre-reform time trends indicated a relative decrease in dying at home (aRRR 0.97, 95% CI 0.97–0.98), at hospital (aRRR 0.96, 95% CI 0.95–0.96) and elsewhere (aRRR 0.94, 95% CI 0.93–0.96) compared to NH for persons without dementia (Fig 1, S2 Table). After the reform, there was no evidence of a step change in dying at home (aRRR 1.08, 95% CI 0.99–1.17), but there was evidence of a significant negative step change for dying at hospital (aRRR 0.83, 95% CI 0.78–0.88) and elsewhere (aRRR 0.65, 95% CI 0.56–0.75) compared to NH deaths followed by a small but significant acceleration in the aRRR for hospital deaths (aRRR 1.02, 95% CI 1.01–1.03) and elsewhere (aRRR 1.07, 95% CI 1.05–1.09) in recent years compared to previous years. Males were more likely than females to die at home (aRRR 1.81, 95%CI 1.77–1.85), in hospital (aRRR 1.62, 95%CI 1.59–1.64) or in another location (aRRR 2.49, 95%CI 2.40–2.59) than in a NH.

**Persons with dementia.** Before the introduction of the reform, for persons with dementia there was significant decrease in hospital deaths (aRRR 0.93, 95%CI 0.91–0.96) compared to NH deaths. After the reform, there was evidence of a step change in the proportions of persons with dementia dying at home (aRRR 2.88, 95%CI 2.13–3.90) versus NH deaths followed by a significant deceleration in home deaths (aRRR 0.90, 95% CI 0.86–0.94) and a small but significant acceleration in hospital deaths (aRRR 1.04, 95% CI 1.00–1.08) in recent years compared to previous years was also observed (S2 Table). Results indicate a substantial relative proportional change in hospital deaths; however, in absolute terms, this represents few older adults due to the small population size dying in hospital (Fig 1). Males with dementia were less likely than females to die at home (aRRR 0.87, 95% CI 0.80–0.95) and were more likely to die in hospital (aRRR 2.08, 95% CI 1.96–2.21).

## Discussion

### Main findings

The number of people dying in hospital decreased since the 2012 reform for the total population, for persons with and without dementia, while NH deaths increased. Immediately after the reform, home deaths increased for persons with dementia but returned to pre-reform levels over time.

Fewer hospital deaths could be a consequence of the Coordination Reform enabling greater collaboration between NH medicine and palliative care in NHs [22, 23]. After the collaboration, Norwegian NHs were better equipped to handle end-of-life palliative care, resulting in fewer transfers to hospitals in the last weeks of life. The 2009 Coordination Reform white paper recommended municipalities to increase the number of palliative units in NHs [1]. A report in 2017 found substantial increases in palliative units and beds indicating that the Coordination Reform contributed to palliative care provisions in NHs [24]. Furthermore, there were already trends in a decreased number of deaths in Norwegian hospital pre-reform [13]. A Norwegian study that evaluated location of death in Norway over 25 years (1987–2011 period) found shifts in end-of-life care from hospital to NHs [13]. The authors concluded that this was partly due to policy shifts enabling NHs to provide end-of life care [13]. Our study extends this literature by evaluating the effects of the 2012 reform which appear to have increased the magnitude and reinforced previous policy reform to avoid hospital deaths. Previous studies found transfers to hospitals and death in hospital was negatively associated with quality of life for older adults and persons with dementia [15–18, 25].

Despite past studies finding death at home being the primary preference [26–28] and policy goals to enable home deaths, we found that there has not been a change in home deaths over time. Kjellstadli et al. [29] found in a population-based, longitudinal analyses, that general practitioner (GP) home visit(s) and interdisciplinary collaboration(s) in the last 3 months before death, significantly increased the odds of dying at home in a dose-dependent manner. However, only a minority (less than 10%) utilized both these GP services in the last month of life. Kjellstadli et al. recommended [29] greater utilization of GPs and primary care to deliver end of life care. Furthermore, interventions to increase awareness, support and education in homecare services are needed to enable more persons to die at home. Recent work found trajectories of home nursing hours and probability of short-term NH stays indicated possible effective palliative home nursing for some, while others, had not accessed services for staying at home longer at the end-of-life [30]. The authors concluded that continuity of care was an important factor in palliative home care and home death [30].

Although overall proportions of home deaths may not have changed, there is evidence that time spent in the community has increased in Norway. Previous literature also found length of stay in Norwegian long-term NHs has decreased since the 2012 reform, to a median of 1.31 years in 2016 [31]. One study of 47 Norwegian NHs conducted in 2012–2014 (n = 691 patients) found 25% of patients died within one year of NH admission [32].

### International comparisons

A European Commission report [33] that compared reforms to long-term care provisions in 35 countries in the past 10 years (2008–2018) found three overall trends: 1) changes to the long-term care policy mix and shifts from residential care towards home care and community care, 2) improving monetary sustainability and 3) increasing access and affordability of care, including recognizing the importance of informal caregivers. Contrary to European trends, Norway has attempted to increase the quality of care in long-term care to avoid hospital deaths. The UK [34], Belgium [35] and Germany [36] have also tried to shift deaths from

hospital to long-term care. Norway like other European countries, is interested in financial sustainability, and has also invested in homecare and the community care.

Very few studies evaluate the effects of long-term care reforms on location of death which is surprising considering healthcare resources appear to play a greater role in location of death than individual-level characteristics [37–40]. Gao et al. [34, 38] evaluated the United Kingdom's National End of Life Care Program [41] aimed to decrease unnecessary emergency admissions, reduce hospital death, improve the skills of the workforce and enable more people to die at the place of their choice [42, 43] and found a decrease in hospital deaths and an increase of home deaths for cancer patients since the care implementation. Gao et al. [39] also proposed a population-level framework to evaluate health services and location of death using health services characteristics and patient-level factors.

Location of death for persons with dementia varies across Europe [44]. However, most persons with dementia die in a long-term care facility [44]. To our knowledge, there are no published European studies evaluating the association of national long-term care reforms and location of death for persons with dementia. There have been national plans created in Denmark and Greece targeting care for persons with dementia, but it is unknown if they are associated with a change of location of death [33].

## Strengths and limitations

Strengths of this study include being the first study to evaluate effect of the Coordination Reform on place of death, using high quality, longitudinal registry data for the whole population. Previous studies have evaluated place of death of people dying from dementia from an international perspective; however, they were cross-sectional and did not focus on countries that have made policy reform [44]. However, there remains several limitations. First, using death certificate data that does not provide detailed information regarding changes in places of care closest to death. Second, we relied on the death certificate to identify persons with dementia making these estimates a conservative underestimate [45]. We know that at least 80% of persons in Norwegian long-term NHs have cognitive impairment [46]. There may be other sociodemographic and health factors that are related to the place of death, but those data were not available. Third, we present population-level results that are not person-specific; however, the strength of this study is that the results apply to the entire Norwegian population.

## Health policy implications and generalizability

This study contributes to society by providing new information on how current strategies have changed end-of-life care for the total population and persons with dementia over time. By evaluating existing care frameworks, we can better understand what is effective based on countries that have actively targeted in-home services supporting older adults to live in the community. At an international level, more research is required to evaluate long-term reforms to create evidenced-based health policy. This research may provide a strategic policy roadmap for countries to follow. Despite our results showing modest change in location of death, these reforms can be considered a success as they enabled treatment in place and created societal awareness in advance care planning. Furthermore, there was economic benefit because care was provided closer to home or in a nursing home and avoided stressful end of life hospital admissions [47]. From a clinical perspective, during the SARS-COV-2 outbreak, we clearly saw the benefits of the Coordination Reform because the nursing homes had previously scaled up the medical staff and had the comprehensive training to provide end of life care [47]. These approaches will be valuable for future investigation for the impact of SARS-COV-2. As all datasets used were national registry data, the generalizability of these results is robust.

## Conclusion

This study provides preliminary evidence at a population-level that the 2012 Norwegian reform enabled treatment in place because of increased older adults having their location of death in a long-term care facility instead of a hospital regardless of dementia status. The number of people dying at home did not change irrespective of patient population group.

## Supporting information

**S1 Table. STROBE statement guidelines.**
(DOCX)

**S2 Table. Adjusted relative risk ratios from multinomial logistic regressions of location of death for the total population and by dementia status weighted to adjust for population growth over the study period (2012–2017).** aRRR = Adjusted Relative Risk Ratio. Adjusted Relative Risk ratios from three multinomial logistic regression are presented where the reference group is long-term care home. Each regression was adjusted for sex and seasonality.
(DOCX)

**S1 File. Aggregated dataset.**
(XLS)

## Author Contributions

**Conceptualization:** Janet L. MacNeil Vroomen, Heather G. Allore, Jenny T. van der Steen, Bettina Husebo.

**Data curation:** Camilla Kjellstadli, Bettina Husebo.

**Formal analysis:** Janet L. MacNeil Vroomen, Camilla Kjellstadli.

**Funding acquisition:** Janet L. MacNeil Vroomen, Heather G. Allore, Jenny T. van der Steen, Bettina Husebo.

**Investigation:** Janet L. MacNeil Vroomen, Camilla Kjellstadli, Heather G. Allore, Bettina Husebo.

**Methodology:** Janet L. MacNeil Vroomen, Heather G. Allore.

**Project administration:** Janet L. MacNeil Vroomen, Camilla Kjellstadli, Bettina Husebo.

**Resources:** Janet L. MacNeil Vroomen, Camilla Kjellstadli, Bettina Husebo.

**Software:** Janet L. MacNeil Vroomen.

**Supervision:** Janet L. MacNeil Vroomen, Camilla Kjellstadli, Heather G. Allore, Jenny T. van der Steen, Bettina Husebo.

**Validation:** Janet L. MacNeil Vroomen.

**Visualization:** Janet L. MacNeil Vroomen, Camilla Kjellstadli, Heather G. Allore, Jenny T. van der Steen.

**Writing – original draft:** Janet L. MacNeil Vroomen, Camilla Kjellstadli, Jenny T. van der Steen, Bettina Husebo.

**Writing – review & editing:** Janet L. MacNeil Vroomen, Camilla Kjellstadli, Heather G. Allore, Jenny T. van der Steen, Bettina Husebo.

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
