## [Decision Letter · Decision Letter 0]

3 Aug 2020

PONE-D-20-19299

Reform influences location of death: Interrupted time-series analysis on older adults and persons with dementia

PLOS ONE

Dear Dr. MacNeil Vroomen,

Thank you for submitting your manuscript to PLOS ONE. After careful consideration by 3 Reviewers and an Academic Editor, all of the critiques of all three Reviewers, especially Reviewer #1, must be addressed in detail in a revision to determine publication status. If you are prepared to undertake the work required, I would be pleased to reconsider my decision, but revision of the original submission without directly addressing the critiques of the three Reviewers does not guarantee acceptance for publication in PLOS ONE. If the authors do not feel that the queries can be addressed, please consider submitting to another publication medium. A revised submission will be sent out for re-review. The authors are urged to have the manuscript given a hard copyedit for syntax and grammar.

**Comments to the Author**

1. Is the manuscript technically sound, and do the data support the conclusions?

Reviewer #1: Partly

Reviewer #2: Yes

Reviewer #3: Yes

2. Has the statistical analysis been performed appropriately and rigorously? 

Reviewer #1: Yes

Reviewer #2: Yes

Reviewer #3: Yes

3. Have the authors made all data underlying the findings in their manuscript fully available?

Reviewer #1: Yes

Reviewer #2: Yes

Reviewer #3: Yes

4. Is the manuscript presented in an intelligible fashion and written in standard English?

Reviewer #1: Yes

Reviewer #2: Yes

Reviewer #3: Yes

5. Review Comments to the Author

Reviewer #1: “Reform influences location of death: Interrupted time-series analysis on older adults and persons with dementia” examines the effect of a national reform on place of death for the general population and stratified by dementia diagnosis (yes/no). The idea is important, the use of population level data is strong, and the methods are novel. Overall, the study shows that the reform did not move the needle on home death very much, but that an existing trend to push death from the hospital to nursing homes continued after the reform was enacted.

Major comments

1. From the introduction, analysis, and discussion, there is not a compelling case that the 2012 reform is a policy cut point that makes sense. Troubling aspects are:

a. Concurrent enactment of other reforms to support end-of-life care in nursing homes seems to undermine the potential impact of the 2012 reform on its own.

b. The reform was enacted in 2012, but surely the municipalities were not able to respond to the reform on January 1, 2012? Or was there lead time given to allow municipalities to adjust local resources in response to the reform?

c. Provide additional support for why January 2012 is a good—and not just convenient—cut point, or how results may be affected by delayed implementation/response to reforms at regional levels. This is particularly important given:

i. Trends observed prior to the 2012 reform appear to have just continued, and in some cases reversed, after the reform.

ii. There were other parallel reforms at/around the same time that could explain the increase in nursing home deaths, which would naturally come from hospital deaths given that is the second most frequent place of death.

d. Provide a clearer explanation of what authors can and cannot say about the 2012 reform based on the analysis performed.

2. Explain in the introduction how hospital death is worse for persons with dementia. It is stated in the abstract and cited in the discussion, but not explained or cited in the introduction, making it difficult to understand why the analysis was stratified based on dementia diagnosis.

3. The use and representation of predicted probabilities needs to be clarified. The statistical analysis refers to calculation of predicted probabilities.

a. I assume the Figure graphs predicted probabilities but the Y axis units are not labeled and the text refers to “adjusted proportions.” Predicted probabilities are not proportions.

b. Clarify how predicted probabilities were calculated: were variables taken at the mean? Mode? Label figure and update text accordingly.

Minor comments

4. It is unclear how deaths in short- versus long-term nursing homes factor into the analysis. This is mentioned in the hypotheses but not followed up in the analysis.

5. Reform/Intervention variable is explained twice on page 5 under “explanatory variables” and “statistical analysis.”

6. For persons not familiar with the methods used, explain what accelerations and decelerations mean. Specifically, do they indicate that any initial effects from the reform were not lasting?

7. The figure is illegible and too wide. Perhaps collapse time points to report on yearly figures as seasonality is adjusted for but not discussed and therefore does not seem that central to the authors’ argument.

8. Typo in S2 Appendix: should 2015 reform read 2012 reform?

Reviewer #2: A repeated cross-sectional, population-level study using mortality data from the Norwegian Cause of Death Registry using an interrupted time series analysis was appropriate to describe changes in the proportion of older adults with and without dementia dying in nursing homes, home, hospital and other locations since Norway instituted a Coordination Reform. This study was developed using the STROBE and RECORD statement guidelines. The statistical analysis was completed appropriately by including a weighted multinomial logistic regression. All data is available publicly.

Reviewer #3: The manuscript presented a clear and well-conducted study on an important topic for EOL care. Its methods and analyzes are adequate to the proposed objectives. My suggestions are: reduce the Introduction, leaving the second paragraph for Discussion; and, in the Main findings, a broader comparison with other countries is necessary to allow some generalization of the results. No further inconsistances were found.

6. PLOS authors have the option to publish the peer review history of their article (what does this mean?). If published, this will include your full peer review and any attached files.

**Do you want your identity to be public for this peer review?** For information about this choice, including consent withdrawal, please see our Privacy Policy.

Reviewer #1: No

Reviewer #2: **Yes: **Rebecca Anne McEwen

Reviewer #3: **Yes: **Fernando Cesar Iwamoto Marcucci

We look forward to receiving your revised manuscript.

Kind regards,

Stephen D. Ginsberg, Ph.D.

Section Editor

PLOS ONE

2.We note that you have indicated that data from this study are available upon request. PLOS only allows data to be available upon request if there are legal or ethical restrictions on sharing data publicly. For information on unacceptable data access restrictions, please see http://journals.plos.org/plosone/s/data-availability#loc-unacceptable-data-access-restrictions.

3. Please upload a copy of Supporting Information Table S1 which you refer to in your text on page 9.

---

## [Author Response · Author response to Decision Letter 0]

21 Aug 2020

To the Section Editor of PLOS ONE

Stephen Ginsberg, PhD 

Amsterdam, August 21, 2020

Dear Dr. Ginsberg,

Thank you for allowing us to revise our manuscript (Ref. No.: PONE-D-20-19299). We thank the reviewers and Editor for their constructive criticism. Below each comment we start our reply with “ANSWER” and in italics we present the changes made in the manuscript along with the page number.

Response to Reviewer

Reviewer #1: “Reform influences location of death: Interrupted time-series analysis on older adults and persons with dementia” examines the effect of a national reform on place of death for the general population and stratified by dementia diagnosis (yes/no). The idea is important, the use of population level data is strong, and the methods are novel. Overall, the study shows that the reform did not move the needle on home death very much, but that an existing trend to push death from the hospital to nursing homes continued after the reform was enacted.

Major comments

1. From the introduction, analysis, and discussion, there is not a compelling case that the 2012 reform is a policy cut point that makes sense. 

ANSWER: The Coordination Reform was introduced in a white paper [1] in 2009 and municipalities were thoroughly prepared before it was enacted in 2012. In fact, there were economic sanctions in place for municipalities who were unprepared [2]. 

[page 3, Introduction, line 50-52]

“After the Coordination Reform was introduced in a white paper in 2009, the Norwegian government implemented it in 2012 as a response to increasing costs, to ensure sustainability of the health care system [1].” 

[page 6, Methods, statistical analysis, line 134-136]

“Lag variables were not created because the policy was enacted on January 1, 2012 after 3 years of notice. There were economic sanctions for municipalities who were unprepared before January 1, 2012 [1, 4].”

Troubling aspects are:

a. Concurrent enactment of other reforms to support end-of-life care in nursing homes seems to undermine the potential impact of the 2012 reform on its own.

ANSWER: We revised our wording to state:

[page 10, Discussion, Main findings, line 183-193]

“Fewer hospital deaths could be a consequence of the Coordination Reform enabling greater collaboration between NH medicine and palliative care in NHs [22, 23]. After the collaboration, Norwegian NHs were better equipped to handle end-of-life palliative care, resulting in fewer transfers to hospitals in the last weeks of life. Furthermore, there were already trends in a decreased number of deaths in Norwegian hospital pre-reform [13]. A Norwegian study that evaluated location of death in Norway over 25 years (1987–2011 period) found shifts in end-of-life care from hospital to NHs [13]. The authors concluded that this was partly due to policy shifts enabling NHs to provide end-of life care [13]. Our study extends this literature by evaluating the effects of the 2012 reform which appear to have increased the magnitude and reinforced previous policy reform to avoid hospital deaths. Previous studies found transfers to hospitals and death in hospital was negatively associated with quality of life for older adults and persons with dementia [15-18, 24].” 

The article that we cited from Kalseth reported reforms to hospital in 1997. Our approach to evaluate the Coordination Reform remains valid even in the face of other medical, social and policy changes over the time period. We are not attempting to partition all the variation to different factors, rather to associate the Coordination Reform with the outcome. Also, there were no other major health care reforms in the time period we are investigating. 

b. The reform was enacted in 2012, but surely the municipalities were not able to respond to the reform on January 1, 2012? Or was there lead time given to allow municipalities to adjust local resources in response to the reform?

ANSWER: Please see the answer to comment #1.

c. Provide additional support for why January 2012 is a good—and not just convenient—cut point, or how results may be affected by delayed implementation/response to reforms at regional levels. This is particularly important given:

ANSWER: Please see answer to comment # 1. 

i. Trends observed prior to the 2012 reform appear to have just continued, and in some cases reversed, after the reform.

ANSWER: As we model pre-reform trends, the moment of the reform and after, we can conclude that there are statistical differences after the implementation of the reforms compared to before the reform. The aim of this study is to evaluate changes in location over time; however, a major question when looking at health policy evidence in evaluations is the magnitude of the reform effect [3]. Our results show that there was a small increase in the magnitude of nursing home deaths compared to the preform trend in addition to a decrease in hospital deaths. Location of home deaths did not really change over time. However these reforms can be considered a success as they created societal awareness in advance care planning and there was economic benefit because care was provided closer to home or in a nursing home and avoided stressful end of life hospital admissions [4]. During the SARS-COV-2 we clearly saw the benefits of the Coordination Reform because the nursing homes had previously scaled up the medical staff and had the training to provide end of life care [4]. Our results are important as they provide evidence from the effect of the Coordination Reform over time.

[page 11, Discussion, Health policy implications, line 233-240]

“Despite our results showing modest change in location of death these reforms can be considered a success as they enabled treatment in place and created societal awareness in advance care planning. Furthermore, there was economic benefit because care was provided closer to home or in a nursing home and avoided stressful end of life hospital admissions [35]. From a clinical perspective, during the SARS-COV-2 outbreak, we clearly saw the benefits of the Coordination Reform because the nursing homes had previously scaled up the medical staff and had the comprehensive training to provide end of life care [35]. These analytics approaches will be valuable for future investigation for the impact of SARS-COV-2.”

ii. There were other parallel reforms at/around the same time that could explain the increase in nursing home deaths, which would naturally come from hospital deaths given that is the second most frequent place of death.

ANSWER: Please see answer to comment #1a.

d. Provide a clearer explanation of what authors can and cannot say about the 2012 reform based on the analysis performed.

ANSWER: For the total population and persons with dementia, it appears the Coordination Reform was associated with an increased aRRR of people dying in the community. This is supported by more people dying in long-term care as well as dying at home and later in the slope change, an increased risk of dying in hospital. We have also added to the limitations section that we present population-level results that are not person specific. The strength of this study is that the results apply to the complete Norwegian population.

[page 11, Discussion, Conclusion, line 242-245]

“This study provides preliminary evidence at a population-level that the 2012 Norwegian reform enabled treatment in place because of increased older adults having their location of death in a long-term care facility instead of a hospital regardless of dementia status. The number of people dying at home did not change irrespective of patient population group.”

[page 10, Discussion, Limitations, line 221-222]

“Third, we present population-level results that are not person-specific; however, the strength of this study is that the results apply to the entire Norwegian population.”

2. Explain in the introduction how hospital death is worse for persons with dementia. It is stated in the abstract and cited in the discussion, but not explained or cited in the introduction, making it difficult to understand why the analysis was stratified based on dementia diagnosis.

ANSWER:

[page 3, Introduction, line 75-79]

“It is also unknown if the Coordination Reform is associated with more persons with dementia dying in hospital. Persons with dementia admitted to hospital are at risk for functional decline, lack of pain control, increased morbidity, increased mortality and a decreased quality of life [15-17]. Furthermore, numerous nursing and medical procedures may be unnecessarily continued or started in the last hours of a patient with dementia’s life [18] and persons with dementia are at high risk for delirium [19].” 

3. The use and representation of predicted probabilities needs to be clarified. The statistical analysis refers to calculation of predicted probabilities.

ANSWER: After running the multinomial logistic regressions, we calculated the year-specific mean predicted probability of dying in at home, nursing home, hospital and other locations for the total population and by dementia status. We then collapsed the predicted probabilities by year and plotted the values over time. 

The revised text reads [page 5, Methods, statistical analysis, line 125-127]:

Three weighted multinomial logistic regressions were performed for the total population, persons with dementia and without to calculate adjusted relative risk ratios (aRRR) and year-specific mean predicted probabilities of location of death.

a. I assume the Figure graphs predicted probabilities but the Y axis units are not labeled and the text refers to “adjusted proportions.” Predicted probabilities are not proportions.

ANSWER: Thank you, we have revised the figure and text to read

Three weighted multinomial logistic regressions were performed for the total population, persons with dementia and without to calculate adjusted relative risk ratios (aRRR) and year-specific mean predicted probabilities of location of death.

b. Clarify how predicted probabilities were calculated: were variables taken at the mean? Mode? Label figure and update text accordingly.

ANSWER: By using the margins command in Stata, we estimated the marginal (e.g. mean over the observed time period) predicted probabilities.

Minor comments

4. It is unclear how deaths in short- versus long-term nursing homes factor into the analysis. This is mentioned in the hypotheses but not followed up in the analysis.

ANSWER: We deleted this terminology throughout the manuscript.

5. Reform/Intervention variable is explained twice on page 5 under “explanatory variables” and “statistical analysis.”

ANSWER: This was deleted from the statistical analysis.

6. For persons not familiar with the methods used, explain what accelerations and decelerations mean. Specifically, do they indicate that any initial effects from the reform were not lasting?

ANSWER: This means that the slope increased or decreased and is now added at the first usage.

[page 8, Results, multinomial logistic regression, line 151-155]:

This was followed by a small but significant deceleration (e.g. reduced slope) in home deaths (aRRR 0.98, 95%CI 0.96-0.98) and a similarly small but significant acceleration (e.g. increased slope) in hospital deaths (aRRR 1.02, 95%CI 1.01-1.02) and dying elsewhere (aRRR 1.06, 95%CI 1.05-1.08) in recent years compared to previous years.

7. The figure is illegible and too wide. Perhaps collapse time points to report on yearly figures as seasonality is adjusted for but not discussed and therefore does not seem that central to the authors’ argument.

ANSWER: We revised the figure

8. Typo in S2 Appendix: should 2015 reform read 2012 reform?

ANSWER: This has been changed.

Reviewer #2: A repeated cross-sectional, population-level study using mortality data from the Norwegian Cause of Death Registry using an interrupted time series analysis was appropriate to describe changes in the proportion of older adults with and without dementia dying in nursing homes, home, hospital and other locations since Norway instituted a Coordination Reform. This study was developed using the STROBE and RECORD statement guidelines. The statistical analysis was completed appropriately by including a weighted multinomial logistic regression. All data is available publicly.

We thank the reviewer for their time to review the manuscript.

Reviewer #3: The manuscript presented a clear and well-conducted study on an important topic for EOL care. Its methods and analyzes are adequate to the proposed objectives. My suggestions are: reduce the Introduction, leaving the second paragraph for Discussion; and, in the Main findings, a broader comparison with other countries is necessary to allow some generalization of the results. No further inconsistances were found.

ANSWER: We thank the reviewer for their time to review the manuscript. We kindly disagree with moving the second paragraph because it makes the case for the evidence gap. We agree with the reviewer that it would be improve the manuscript to make comparisons with other countries implementing reforms to their long-term care. Unfortunately although reforms are frequently happening to long-term care as a response to the greying of the population, it is rare to make an evaluation of the reform. We rewrote the Health policy implications and generalizability to increase the external validity of the paper and to encourage other countries to create other evidenced-based health policy.

[page 10, Discussion, Health policy implications and generalizability, line 224-240]:

 “This study contributes to society by providing new information on how current strategies have changed end-of-life care for the total population and persons with dementia over time. By evaluating existing care frameworks, we can better understand what is effective based on countries that have actively targeted in-home services supporting older adults to live in the community. In contrast to Norway investing in the community medical infrastructure, the Netherlands implemented austerity measures that included closing down long-term care facilities [35] and cutting homecare by 32% [36]. It is unclear if the Netherlands has overlooked the merits of home care and long-term care facilities, particularly for people with dementia. At an international level, more research is required to evaluate long-term reforms to create evidenced-based health policy. This research may provide a strategic policy roadmap for countries to follow. Despite our results showing modest change in location of death these reforms can be considered a success as they enabled treatment in place and created societal awareness in advance care planning. Furthermore, there was economic benefit because care was provided closer to home or in a nursing home and avoided stressful end of life hospital admissions [37]. From a clinical perspective, during the SARS-COV-2 outbreak, we clearly saw the benefits of the Coordination Reform because the nursing homes had previously scaled up the medical staff and had the comprehensive training to provide end of life care [37]. These approaches will be valuable for future investigation for the impact of SARS-COV-2. As all datasets used were national registry data, the generalizability of these results are robust.”

References

1. Ministry of health and care services. Samhandlingsreformen. Rett behandling - på rett sted - til rett tid [The Coordination Reform. Proper treatment – at the right place and right time]. Oslo: Ministry of health and care services, 2009 Report Meld.st.47 (2008-2009).

2. Forskningsrådet. Evaluering av samhandlingsreformen. Sluttrapport fra styringsgruppen for forskningsbasert følgeevaluering av samhandlinvsreformen (EVASAM). Oslo: 2016.

3. Baicker K, Chandra A. Evidence-Based Health Policy. New England Journal of Medicine. 2017;377(25):2413-5. doi: 10.1056/NEJMp1709816. PubMed PMID: 29262287.

4. Husebø BS, Berge LI. Intensive Medicine and Nursing Home Care in Times of SARS CoV-2: A Norwegian Perspective. The American journal of geriatric psychiatry : official journal of the American Association for Geriatric Psychiatry. 2020;28(7):792-3. Epub 04/22. doi: 10.1016/j.jagp.2020.04.016. PubMed PMID: 32381282.

---

## [Decision Letter · Decision Letter 1]

7 Sep 2020

PONE-D-20-19299R1

Reform influences location of death: Interrupted time-series analysis on older adults and persons with dementia

PLOS ONE

Dear Dr. MacNeil Vroomen,

Thank you for resubmitting your manuscript to PLOS ONE. Please address the minor concerns of Reviewers #1 and #3 so I can render a decision on the manuscript.

**Comments to the Author**

1. If the authors have adequately addressed your comments raised in a previous round of review and you feel that this manuscript is now acceptable for publication, you may indicate that here to bypass the “Comments to the Author” section, enter your conflict of interest statement in the “Confidential to Editor” section, and submit your "Accept" recommendation.

Reviewer #1: (No Response)

Reviewer #2: All comments have been addressed

Reviewer #3: All comments have been addressed

2. Is the manuscript technically sound, and do the data support the conclusions?

Reviewer #1: Yes

Reviewer #2: Yes

Reviewer #3: Yes

3. Has the statistical analysis been performed appropriately and rigorously? 

Reviewer #1: Yes

Reviewer #2: Yes

Reviewer #3: Yes

4. Have the authors made all data underlying the findings in their manuscript fully available?

Reviewer #1: Yes

Reviewer #2: Yes

Reviewer #3: Yes

5. Is the manuscript presented in an intelligible fashion and written in standard English?

Reviewer #1: Yes

Reviewer #2: Yes

Reviewer #3: Yes

6. Review Comments to the Author

Reviewer #1: The authors addressed concerns raised in the initial review by providing additional explanation of the context in which reforms existed and by clarifying analyses. Two minor comments:

1. Clarify in statistical analysis that, when calculating predicted probabilities, all other variables were held at their means.

2. Change "was" to "were" in the first sentence under "explanatory variables."

Reviewer #2: The authors provided strong supporting evidence to address the concerns raised by the reviewer. All data was supplied.

Reviewer #3: The manuscript explores an important topic in EOL care, and the method and data analysis are clear.

In Results, as seasonality and sex were included as covariate, and nothing else was explained about that, I assume that they didn´t show differences, but it could be briefly declared in the text. I still missed an international approach in Discussion, many countries (in Europe, to be close to the manuscript context) presented a progressive change in place of death from hospital to nursing homes and to home in the last decades, and how much it can be explained by public health policies, epidemiological factors or by wider access to palliative care approach. In particular on Coordination Reform, I would like to ask whether it included specific topics on palliative care in the document, as it is not familiar to most readers.

7. PLOS authors have the option to publish the peer review history of their article (what does this mean?). If published, this will include your full peer review and any attached files.

**Do you want your identity to be public for this peer review?** For information about this choice, including consent withdrawal, please see our Privacy Policy.

Reviewer #1: No

Reviewer #2: **Yes: **Rebecca McEwen

Reviewer #3: No

We look forward to receiving your revised manuscript.

Kind regards,

Stephen D. Ginsberg, Ph.D.

Section Editor

PLOS ONE

---

## [Author Response · Author response to Decision Letter 1]

6 Oct 2020

Dear Editor and Reviewers,

Thank you for allowing us to revise our manuscript “Reform influences location of death: Interrupted time-series analysis on older adults and persons with dementia” (Ref. No.: PONE-D-20-19299R1). We thank the reviewers and Editor for their constructive criticism. Below each comment we start our reply with “ANSWER” and in italics we present the changes made in the manuscript along with the page number.

Response to Reviewer

Reviewer #1: The authors addressed concerns raised in the initial review by providing additional explanation of the context in which reforms existed and by clarifying analyses. 

Two minor comments:

1. Clarify in statistical analysis that, when calculating predicted probabilities, all other variables were held at their means.

ANSWER: Thank you, we clarified and updated the methods section.

[Statistical Analysis section, page 5, lines:127-128] 

"When calculating predicted probabilities, all other variables were held at their means.”

2. Change "was" to "were" in the first sentence under "explanatory variables."

ANSWER: We corrected the typo.

[Statistical Analysis section, page 5, lines:117-118] 

"A time variable (in cumulative quarters) and policy dummy variables indicating the pre-intervention period (coded 0) or the post-intervention period (coded 1) were created.”

Reviewer #3:

1. The manuscript explores an important topic in EOL care, and the method and data analysis are clear.

In Results, as seasonality and sex were included as covariate, and nothing else was explained about that, I assume that they didn´t show differences, but it could be briefly declared in the text.

ANSWER: We were only using seasonality and sex variables as covariates to answer our main research question and therefore we did not report their results. Guidelines to evaluate healthcare services on location of death [1] recommend including them as variables to be controlled for when evaluating the service impact however they are not the focus of the evaluation. However, we can understand readers may be interested in these variables so we added the seasonality and sex parameter estimates to the table in the appendix. Futhermore, we added text in the results section. For the total population and persons without dementia, males were more likely than females to die at home, in hospital or in another location than a long-term care facility. However males with dementia were less likely than females to die at home and were more likely to die in hospital. Calendar years were not significant in any of the regressions.

[Results section, total population, page 8, lines:156-157] 

“Males were more likely than females to die at home (aRRR 1.86, 95%CI 1.82-1.90), in hospital (aRRR 1.75, 95%CI 1.73-1.78) or in another location (aRRR 2.53, 95%CI 2.44-2.62) than in a NH.”

[Results section, Persons without dementia, page 8, lines:170-172] 

“Males were more likely than females to die at home (aRRR 1.81, 95%CI 1.77-1.85), in hospital (aRRR 1.62, 95%CI 1.59-1.64) or in another location (aRRR 2.49, 95%CI 2.40-2.59) than in a NH.”

[Results section, Persons with dementia, page 9, lines:181-183 

“Males with dementia were less likely than females to die at home (aRRR 0.87, 95% CI 0.80-0.95) and were more likely to die in hospital (aRRR 2.08, 95% CI 1.96-2.21).”

2. I still missed an international approach in Discussion, many countries (in Europe, to be close to the manuscript context) presented a progressive change in place of death from hospital to nursing homes and to home in the last decades, and how much it can be explained by public health policies, epidemiological factors or by wider access to palliative care approach.

ANSWER: We have included a new section “International comparisons“.

[Discussion section, International comparisons, page 10, lines: 220-241] 

“International comparisons

A European Commission report [33] that compared reforms to long-term care provisions in 35 countries in the past 10 years (2008-2018) found three overall trends: 1) changes to the long-term care policy mix and shifts from residential care towards home care and community care, 2) improving monetary sustainability and 3) increasing access and affordability of care, including recognizing the importance of informal caregivers. Contrary to European trends, Norway has attempted to increase the quality of care in long-term care to avoid hospital deaths. The UK [34], Belgium [36] and Germany [37] have also tried to shift deaths from hospital to long-term care. Norway like other European countries, is interested in financial sustainability, and has also invested in homecare and the community care. 

Very few studies evaluate the effects of long-term care reforms on location of death which is surprising considering healthcare resources appear to play a greater role in location of death than individual-level characteristics [38-41]. Gao et al. [34, 39] evaluated the United Kingdom’s National End of Life Care Program [42] aimed to decrease unnecessary emergency admissions, reduce hospital death, improve the skills of the workforce and enable more people to die at the place of their choice [35, 43] and found a decrease in hospital deaths and an increase of home deaths for cancer patients since the care implementation. Gao et al. [40] also proposed a population-level framework to evaluate health services and location of death using health services characteristics and patient-level factors. 

Location of death for persons with dementia varies across Europe [44]. However, most persons with dementia die in a long-term care facility [44]. To our knowledge, there are no published European studies evaluating the association of national long-term care reforms and location of death for persons with dementia. There have been national plans created in Denmark and Greece targeting care for persons with dementia, but it is unknown if they are associated with a change of location of death [33].”

3. In particular on Coordination Reform, I would like to ask whether it included specific topics on palliative care in the document, as it is not familiar to most readers.

ANSWER: The coordination reform document written in 2008, contained a chapter about the “Future tasks and the role of municipalities” with a section dedicated to increasing palliative care units in nursing homes. The document listed at the time of publication, 27 palliative units and 164 palliative beds. In 2017 there were 48 palliative units with 294 palliative beds + 147 single palliative beds in Norwegian SNFs.

[Discussion section, Main findings, page 9, lines:192-195] 

“The 2009 Coordination Reform white paper recommended municipalities to increase the number of palliative units in NHs [1]. A report in 2017 found substantial increases in palliative units and beds indicating that the Coordination Reform contributed to palliative care provisions in NHs [24].”

References

1. Gao W, Huque S, Morgan M, Higginson IJ. A Population-Based Conceptual Framework for Evaluating the Role of Healthcare Services in Place of Death. Healthcare (Basel). 2018;6(3). Epub 2018/09/12. doi: 10.3390/healthcare6030107. PubMed PMID: 30200247; PubMed Central PMCID: PMCPMC6164352.

2. Det Kongelige Helse-OG Omsorgsdepartment. Samhandlingsreformen Rett behandling – på rett sted – til rett tid 2009. Available from: https://www.regjeringen.no/contentassets/d4f0e16ad32e4bbd8d8ab5c21445a5dc/no/pdfs/stm200820090047000dddpdfs.pdf.

3. Kaasa S, Andersen S, Bahus M, Broen P, Farsund H, Flovik A, et al. På liv og død. Palliasjon til alvorlig syke og døende [On life and death. Palliative care to the seriously ill and dying]. Oslo: Helse- og omsorgsdepartementet, 2017 Report NOU 2017.

---

## [Editor Report · Decision Letter 2]

9 Oct 2020

Reform influences location of death: Interrupted time-series analysis on older adults and persons with dementia

PONE-D-20-19299R2

Dear Dr. MacNeil Vroomen,

We’re pleased to inform you that your manuscript has been judged scientifically suitable for publication and will be formally accepted for publication once it meets all outstanding technical requirements.

Kind regards,

Stephen D. Ginsberg, Ph.D.

Section Editor

PLOS ONE

---

## [Editor Report · Acceptance letter]

23 Oct 2020

PONE-D-20-19299R2 

Reform influences location of death: Interrupted time-series analysis on older adults and persons with dementia 

Dear Dr. MacNeilVroomen:

I'm pleased to inform you that your manuscript has been deemed suitable for publication in PLOS ONE. Congratulations! Your manuscript is now with our production department. 

Kind regards, 

on behalf of

Dr. Stephen D. Ginsberg 

Section Editor

PLOS ONE